# A Novel Variational Lower Bound For Inverse Reinforcement Learning

## Abstract

Inverse reinforcement learning (IRL) seeks to learn the reward function from expert trajectories, to understand the task for imitation or collaboration thereby removing the need for manual reward engineering. However, IRL in the context of large, high-dimensional problems with unknown dynamics has been particularly challenging. In this paper, we present a new **V**ariational **L**ower **B**ound for **IRL** (VLB-IRL), which is derived under the framework of a probabilistic graphical model with an optimality node. Our method simultaneously learns the reward function and policy under the learned reward function by maximizing the lower bound, which is equivalent to minimizing the reverse Kullback-Leibler divergence between an approximated distribution of optimality given the reward function and the true distribution of optimality given trajectories. This leads to a new IRL method that learns a valid reward function such that the policy under the learned reward achieves expert-level performance on several known domains. Importantly, the method outperforms the existing state-of-the-art IRL algorithms on these domains by demonstrating better reward from the learned policy.

## 1 Introduction

Reinforcement learning (RL) is a popular method for automating decision making and control. However, to achieve practical effectiveness, significant engineering of reward features and reward functions has traditionally been necessary. Recently, the advent of deep reinforcement learning has eased the need for feature engineering for policies and value functions, and demonstrated encouraging outcomes for various complex tasks such as vision-based robotic control and video games like Atari and Minecraft. Despite these advancements, reward engineering continues to be a significant hurdle to the application of reinforcement learning in practical contexts.

The problem at hand is to learn a reward function that explains the task performed by the expert, where the learner only has access to a limited number of expert trajectories and cannot ask for more data. While imitation learning is a popular technique for training agents to mimic expert behavior, conventional methods such as behavior cloning and generative adversarial imitation learning Ho & Ermon (2016) often do not explicitly learn the underlying reward function, which is essential for a deeper understanding of the task. To overcome this limitation, researchers have developed inverse reinforcement learning (IRL) to infer the reward function from expert trajectories. IRL offers several advantages over direct policy imitation, including the ability to analyze imitation learning (IL) algorithms, deduce agents' intentions, and optimize rewards in new environments.

In this paper, we present a new probabilistic graphical model involving both the reward function and optimality as key random variables, which can serve as a framework for representing IRL as a probabilistic inference problem. Within the context provided by this model, our main contributions are:

1. A lower bound on the likelihood of the reward model given the expert trajectories as data, which is derived from first principles, and involves minimizing the reverse Kullback–Leibler divergence between an approximated distribution of optimality given the reward function and the true distribution of optimality given trajectories.
2. A novel IRL algorithm which learns the reward function as variational inference that optimizes the lower bound in domains where the state and action spaces can be continuous.

3. Improved learning performance of the algorithm compared to state of the art techniques as seen through the policy given the learned reward, which is demonstrated on multiple well-known continuous domains of various complexity.

Our novel lower bound can not only serve as a point of departure for the formulation of other principled bounds but also stimulate discussions and design of new IRL methods that utilize such bounds.

## 2 A Novel Variational Lower Bound

Inspired by the significant impact of variational lower bounds such as ELBO in RL, we present a variational lower bound on the likelihood of the reward function. To derive this lower bound from first principles, we first introduce a graphical model to represent the problem in Section 2.1 followed by the derivation of the lower bound in Section 2.2. We discuss the appropriateness of an inherent approximation in Section 2.3. All proof can be found in Appendix B.

### 2.1 A Probabilistic Graphical Model of IRL

Inspired by the graphical model for forward RL as shown in Levine (2018) and a general lack of such modeling of IRL, we aim to fill this gap in this section. Recall that we are now interested in learning the reward function given the state and action trajectories from the expert. Therefore, it is necessary to embed the reward function explicitly in the graphical model, as shown in Figure 1. The reward value $r_t$ is conditioned on state $\mathbf{s_t}$ and action $\mathbf{a_t}$ respectively denoting the state and action in the trajectory. We introduce an optimality random variable $\mathcal{O}_t$, whose value is conditioned on the reward value $r_t$. The optimality follows the definition of Levine (2018) in that the optimality variable is a binary random variable, where $\mathcal{O}_t = 1$ denoting that the observed action at time step $t$ is optimal given the state, and $\mathcal{O}_t = 0$ denotes that it is not optimal. We will discuss the reward function $\mathcal{R}$ in more detail in Section 2.3.

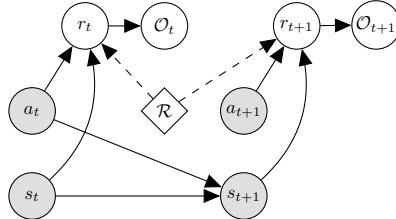
(a) Graphical model with reward function included

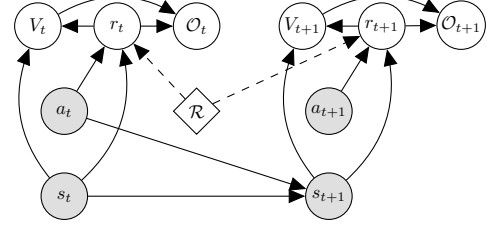
(b) Graphical model with value function included

Figure 1: A probabilistic graphical model for the IRL problem. Shaded nodes represent the observed variables, unshaded nodes represent the latent variables. The dashed line represents the reparameterization trick described later in Section 2.3. Reward value $r_t$ is sampled from distribution $\mathcal{R}(s_t, a_t)$ using reparameterization trick.

The choice of a probabilistic graphical model to represent the relationships between the state, action, reward, and optimality variables in IRL is motivated by the desire to capture the complex dependencies between these variables in a lucid manner. Indeed, the graphical model has a natural property when it comes to analyzing optimality. Specifically, the graphical model represents the conditional independence of the optimality from the state and action variables, given the reward variable. In other words, when given the state-action pair $(s_t, a_t)$, we sum over the reward value $r_t$ to get the marginal conditional distribution of the optimality $\mathcal{O}_t$, that is $p(\mathcal{O}_t|s_t, a_t)$. However, if the reward value $r_t$ is given, then the state-action pair is of course unnecessary in order to get the marginal distribution of the optimality $\mathcal{O}_t$, that is $p(\mathcal{O}_t|r_t)$.

Given the conditional independence in a probabilistic graphical model, which is similar to a Bayesian network, we may write:

$$p(\mathcal{O}_t|r_t, s_t, a_t) = p(\mathcal{O}_t|r_t) \tag{1}$$

which leads to the following marginal:

$$p(\mathcal{O}_t|s_t, a_t) = \int_{r_t} p(\mathcal{O}_t|r_t, s_t, a_t)\, p(r_t|s_t, a_t) = \int_{r_t} p(\mathcal{O}_t|r_t)\, p(r_t|s_t, a_t) \tag{2}$$

## 2.2 VARIATIONAL LOWER BOUND FOR IRL

We use the graphical models of Fig. 1 to formulate the log-likelihood of the observed trajectories. [1]
Here, we are aided by Eqs. 1 and 2 in our derivation, which we detail below.

$$
\begin{aligned}
\log p(\tau) &= \log \int_{\mathcal{O}_t} \int_{r_t} p(\tau, \mathcal{O}_t, r_t) \\
&= \log p(s_1) + \log \prod_t \int_{\mathcal{O}_t} \int_{r_t} p(r_t|s_t, a_t)\, p(\mathcal{O}_t|r_t)\, p(s_{t+1}|s_t, a_t)\, p(a_t) \\
&= \log p(s_1) + \sum_t \log \int_{\mathcal{O}_t} p(\mathcal{O}_t|s_t, a_t)\, p(s_{t+1}|s_t, a_t)\, p(a_t) \quad \text{(using Eq. 2)} \\
&= \log p(s_1) + \sum_t \log \int_{\mathcal{O}_t} \frac{p(\mathcal{O}_t|s_t, a_t)}{q(\mathcal{O}_t|r_t)} q(\mathcal{O}_t|r_t)\, p(s_{t+1}|s_t, a_t)\, p(a_t).
\end{aligned}
\tag{3}
$$

Here, we use variational distribution $q(\mathcal{O}_t|r_t)$ to approximate the true distribution $p(\mathcal{O}_t|s_t, a_t)$. We may think of $q(\mathcal{O}_t|r_t)$ as an approximation of the true distribution of optimality given the trajectories, $p(\mathcal{O}_t|s_t, a_t)$ and we will discuss in more detail in Section 2.3. We may rewrite the equation above by introducing the expectation $\mathbb{E}_{q(\mathcal{O}_t|r_t)}$ and applying Jensen's inequality:

$$
\begin{aligned}
\log p(\tau) &= \log p(s_1) + \sum_t \log \mathbb{E}_{q(\mathcal{O}_t|r_t)} \left[ \frac{p(\mathcal{O}_t|s_t, a_t)}{q(\mathcal{O}_t|r_t)} p(s_{t+1}|s_t, a_t) p(a_t) \right] \\
&\geq \log p(s_1) + \sum_t \mathbb{E}_{q(\mathcal{O}_t|r_t)} \left[ \log \frac{p(\mathcal{O}_t|s_t, a_t)}{q(\mathcal{O}_t|r_t)} + \log p(s_{t+1}|s_t, a_t) + \log p(a_t) \right] \\
&= \log p(s_1) + \sum_t [-\mathrm{KL}(q(\mathcal{O}_t|r_t) \parallel p(\mathcal{O}_t|s_t, a_t)) + \log p(s_{t+1}|s_t, a_t) + \log p(a_t)].
\end{aligned}
\tag{4}
$$

We end up with the following Evidence Lower BOund (ELBO):

$$\mathrm{ELBO} = \log p(s_1) + \sum_t [-\mathrm{KL}(q(\mathcal{O}_t|r_t) \parallel p(\mathcal{O}_t|s_t, a_t)) + \log p(s_{t+1}|s_t, a_t) + \log p(a_t)]. \tag{5}$$

The motivation for using $q(\mathcal{O}_t|r_t)$ to approximate $p(\mathcal{O}_t|s_t, a_t)$ is quite intuitive that both state-action pair and reward value can explain the optimality. An interpretation of $p(\mathcal{O}_t|s_t, a_t)$ is that the probability can be viewed as the confidence with which the state-action pair comes from the expert. Analogously, $q(\mathcal{O}_t|r_t)$ is the confidence with which the reward value $r_t$ belongs to the expert. Additionally, to optimize the reward function, we have to incorporate the reward function into the log-likelihood of trajectories.

Note that if we take the partial derivative of Eq. 5 with respect to $r_t$, we obtain,

$$\frac{\partial\, \mathrm{ELBO}}{\partial\, r_t} = -\sum_t \frac{\partial\, \mathrm{KL}(q(\mathcal{O}_t|r_t) \parallel p(\mathcal{O}_t|s_t, a_t))}{\partial\, r_t} \tag{6}$$

because the remaining terms in equation 4 are constant w.r.t. $r_t$. The derivation from first principles above leads to the following theorem.

**Theorem 1.** *Optimizing the evidence lower bound of the log-likelihood of trajectories w.r.t $r_t$ is equivalent to optimizing the reverse KL divergence between $q(\mathcal{O}_t|r_t)$ and $p(\mathcal{O}_t|s_t, a_t)$ w.r.t $r_t$.*

---

[1]In practice, using full trajectories to estimate the reward function in IRL may lead to high variance especially when the trajectories are long. On the other hand, using single state-action pairs to infer the reward function can lead to more stable and efficient estimates, because each observed pair provides a separate estimate of the reward function that can be combined to reduce variance.

## 2.3 Approximated Distribution of Optimality

Next, we discuss the approximation of the distribution over optimality under the framework of the probabilistic graphical model and offer insights.

Given a state-action pair $(s_t, a_t)$ obtained from some trajectory $\tau$, $p(\mathcal{O}_t|s_t, a_t)$ informs whether the action $a_t$ is optimal at state $s_t$, or in other words, whether the state-action pair comes from the expert's trajectory. This term represents the *true distribution* in the reverse KL divergence contained in the lower bound. To make this classification, we may simply use binary logistic regression, $C_{\boldsymbol{\theta}}$. In algorithms such as GAIL and AIRL, the input to the classifier consists of trajectories from both the expert and the learner. The classifier utilizes these trajectories as input to make predictions about the optimality of the state-action pair.

$$p(\mathcal{O}_t|s_t, a_t) \triangleq C_{\boldsymbol{\theta}}(s_t, a_t) \tag{7}$$

Similarly, given reward value $r_t$, the approximation $q(\mathcal{O}_t|r_t)$ informs whether the reward value leads to optimal behavior, i.e., whether it is induced by the expert or not.

Recall that the reward value $r_t$ is the feedback from the environment. Therefore, we propose this reward value as our first approach to estimate the optimality as defined in Eq. 8.

$$q(\mathcal{O}_t = 1|r_t) \propto e^{r_t}, \quad \text{where } r_t \sim \mathcal{R}(s_t, a_t). \tag{8}$$

Note that the left hand side of Eq. 8 is conditioned on reward value $r_t$, which is distributed according to the distribution $\mathcal{R}(s_t, a_t)$. Thus, we may apply a reparameterization trick to sample a specific reward value $r_t$ without losing track of the gradient propagation under the probabilistic graphical model. The dashed line in Fig. 1 denotes the reparameterization trick and $r_t$ represents the sampled specific reward value from the distribution of $\mathcal{R}(s_t, a_t)$. To illustrate this, consider the simplistic case where the reward value distribution is a univariate Gaussian: $\mathcal{R}(s_t, a_t) = \mathcal{N}(\mu, \sigma^2)$ and let $r_t \sim \mathcal{R}(s_t, a_t)$. In this case, a valid reparameterization is $r_t = \mu + \sigma\epsilon$ (reward values are distributed around the mean), where $\epsilon$ is an auxiliary noise variable, $\epsilon \sim \mathcal{N}(0, 1)$.

However, it is not sufficient to use the reward values to represent the optimality because, in an optimal trajectory, not every state-action pair has the highest reward value at each time step; often we may perform an action that obtains longer term gain while forgoing greedy rewards. Hence, a more robust way of computing the optimality given the distribution over reward value is needed. Here, the *advantage function* is a candidate for the solution:

$$A(s_t, a_t) = Q(s_t, a_t) - V(s_t) = r_{\boldsymbol{\phi}}(s_t, a_t) + \gamma V(s_{t+1}) - V(s_t) \tag{9}$$

In practice, we can use actor-critic-based policy to retrieve the value function estimation, such as PPO, SAC, or TD3. . As the reward value is a component of the advantage function, we can continue to keep track of the reward function. Therefore, the optimality of reward value can be expressed as:

$$q(\mathcal{O}_t = 1|r_t) \propto A(s_t, a_t) = \sigma\left(r_t + \gamma V(s_{t+1}) - V(s_t)\right) \tag{10}$$

where $r_t \sim \mathcal{R}(s_t, a_t)$ and $\sigma$ is the Sigmoid function to normalize the advantage function.

## 2.4 Understanding the Relationship between Two Distributions over Optimality

Notice that the two distributions over optimality $p(\mathcal{O}_t|s_t, a_t)$ and $q(\mathcal{O}_t|r_t)$ have different definitions according to Eqs. 1 and 2. In this subsection, we demonstrate the validity of using $q(\mathcal{O}_t|r_t)$ to approximate $p(\mathcal{O}_t|s_t, a_t)$. We begin with Lemma 1 which establishes an upper bound that is utilized later, and whose proof is in the Appendix.

**Lemma 1.** $|\mathbb{E}[f(X)] - f(\mathbb{E}[X])| \leq M Var(X)$, *where* $|f''(x)| \leq M$ *and* $Var(X)$ *denotes the variance of the distribution of* $X$.

We directly use Lemma 1 to arrive at the theorem below and present its proof in the appendix.

**Theorem 2.** *If* $|p''(\mathcal{O}_t = 1|r_t)| < M$, *then the approximation distribution* $q(\mathcal{O}_t \mid \mathbb{E}[r_t])$, *where* $r_t \sim \mathcal{R}(s_t, a_t)$ *and* $\mathcal{R}(s_t, a_t) = p(r_t|s_t, a_t)$, *approximates the true distribution* $p(\mathcal{O}_t|s_t, a_t)$ *with an approximation error that is bounded by* $M Var[r_t]$.

Note that the approximation error is bounded by the variance of $r_t$ instead of $\mathcal{O}_t$ and the motivation of using reward $r_t$ to approximate state-action pair $(s_t, a_t)$ comes from the Figure.1, in which holds the connectivity that node $r_t$ is connected to node $s_t, a_t$ and node $\mathcal{O}_t$. Without the connectivity, even if the variance of $r_t$ is zero, the approximation error cannot be bounded. More detail can be found in Appendix B.2. It is also the connectivity that holds so that we can derive Eq. 1 and Eq. 2.

From Theorem 2, we know that we can use $q(\mathcal{O}_t \mid \mathbb{E}[r_t])$ to approximate $p(\mathcal{O}_t|s_t, a_t)$ with a bounded approximation error w.r.t $\text{Var}(r_t)$ . Additionally, if the variance of $r_t$ is small enough, then we can use $r_t$ to estimate $\mathbb{E}[r_t]$. Subsequently, the objective function of VLB-IRL is defined in the following:

$$
\begin{aligned}
\mathcal{L}(r_t) &= \sum_t [-\text{KL}(q(\mathcal{O}_t|r_t) \parallel p(\mathcal{O}_t|s_t, a_t))) + \lambda \text{Var}(r_t)] \\
&= \sum_t \left\{ -\text{KL}\left[ \sigma(e^{r_t + \gamma V(s_{t+1}) - V(s_t)}) \parallel C_{\boldsymbol{\theta}}(s_t, a_t) \right] + \lambda \text{Var}(r_t) \right\}
\end{aligned}
\tag{11}
$$

where $\sigma$ is the Sigmoid function to normalize the advantage function and $\lambda$ is the hyperparameter to control the contribution of variance in the objective function.

## 2.5 THE REWARD AMBIGUITY PROBLEM

In this section, we discuss the reward ambiguity problem. As deduced in Ho & Ermon (2016), IRL is a dual of an occupancy measure matching problem and the induced optimal policy is the primal optimum. In the following, we further draw the conclusion that IRL is a dual of the optimality matching problem and the reward function induced by the optimality best explains the expert $\pi^E$.

**Lemma 2.** $p(\mathcal{O}_t|s_t, a_t^*)$ *is maximal if and only if for any given $s_t$, the action $a_t^*$ from the expert among all actions has the highest probability, $p(\mathcal{O}_t = 1|s_t, a_t)$.*

The lemma follows from the definition of the classifier as discussed in Section 2.3, which classifies whether the state-action pairs are from the expert or not.

**Theorem 3.** *The reward function $\mathcal{R}$ best explains the expert policy $\pi^E$ if $p(\mathcal{O}_t|s_t, a_t)$ is maximal and $q(\mathcal{O}_t|r_t)$ is identical to $p(\mathcal{O}_t|s_t, a_t)$, where $r_t \sim \mathcal{R}(s_t, a_t)$.*

From Theorem 3, we are guaranteed to have a reward function $\mathcal{R}$ that best explains the expert policy. Theorem 3 also offers the benefit that we are guaranteed to avoid a degenerate reward function as the solution by optimizing Eq. 6 if $p(\mathcal{O}_t|s_t, a_t)$ is maximal.

## 2.6 THE NOISY EXPERT TRAJECTORIES PROBLEM

Recall that the objective function of VLB-IRL consists of a reverse KL divergence between $q(\mathcal{O}_t|r_t)$ and $p(\mathcal{O}_t|s_t, a_t)$. Compared to the mean-seeking behavior induced by minimizing forward KL divergence, minimizing reverse KL divergence has the behavior that it is mode-seeking, which tends to be more robust to overfitting and can provide better estimates of the true posterior distribution if the posterior distribution is noisy. This is beneficial for IRL because trajectories could be noisy and may contain suboptimal actions, which challenges the learning. By emphasizing the modes of the data distribution, the reverse KL divergence helps identify the most likely explanations for the observed behavior, even in the presence of noise or uncertainty.

Compared to existing state-of-the-art algorithms, most of them use a single neural network to estimate the reward function. However, in VLB-IRL, we use two separate neural networks to update the reward function. The first neural network is the classifier, defined in Eq. 8. The second neural network is the approximation optimality, defined in Eq. 8 and Eq. 10. The architecture of two separate neural networks has a natural property that is robust to overfitting and has better generalization. It is essential for the IRL algorithm with the presence of noise in expert trajectories.

Another major difference is that most prior algorithms focus on learning identical state-action marginal distributions and consequently end up learning the expert's noisy state-action representation as well. However, in VLB-IRL, since the true distribution $p(\mathcal{O}_t|s_t, a_t)$ represents the optimality conditioned on the state-action pair, it has the ability to distinguish and generalize the noisy trajectories.

## 3 VLB-IRL ALGORITHM

We present an algorithm, which we call variational lower bound for IRL, VLB-IRL (Algorithm 1), for optimizing Eq. 11. To do so, we introduce function approximations for $\pi$ and $\mathcal{R}$: we fit a parameterized policy $\pi_\psi$ with weights $\psi$, a parameterized classifier $C_\theta$ with weights $\theta$ and a parameterized reward function $R_\phi$ with weights $\phi$. From line 3 to Line 4, we collect learner policy trajectories and save them into the buffer. From line 7 to line 8, we update the classifier and reward function defined in Section 2.3. At line 9, we update the learner policy based on the learned reward function.

---

**Algorithm 1** VLB-IRL: Variational Lower Bound for Inverse Reinforcement Learning

---

**Require:** Expert trajectories $\tau_i^E$, $N$: # of iterations
 1: Initialize learner policy $\pi_\psi^L$, trajectory buffer $B$, classifier $C_\theta$ and reward function $\mathcal{R}_\phi$
 2: **for** step $t$ in $1, ..., N$ **do**
 3:     Collect learner policy trajectories $\tau_i^L = (s_0, a_0, ..., s_T, a_T)$ by executing $\pi_\psi^L$
 4:     Add trajectories $\tau_i^L$ into trajectory buffer $B$
 5:     Sample random minibatch state-action pairs $(s_t^L, a_t^L)$ from Buffer $B$
 6:     Sample random minibatch state-action pairs $(s_t^E, a_t^E)$ from expert trajectories $\tau_i^E$
 7:     Train classifier $C_\theta$ via binary logistic regression to classify $(s_t^L, a_t^L)$ from $(s_t^E, a_t^E)$
 8:     Update reward function $\mathcal{R}_\phi$ via Equation 11
 9:     Update learner policy $\pi_\psi^L$ with reward function $\mathcal{R}_\phi$
10: **end for**

---

## 4 EXPERIMENTS

To ascertain the efficiency of our method VLB-IRL, we use multiple Mujoco benchmark domains (Todorov et al., 2012), such as LunarLander, Hopper, Walker2d, HalfCheetah, and Ant. We also use another more realistic, robot-centered, and goal-based environment, Assistive Gym (Erickson et al., 2020), to further analyze the performance of VLB-IRL. In our experiments, we compare the average true reward accrued over 50 episodes by our method versus the current state-of-the-art methods such as AIRL (Fu et al., 2018a), $f$-IRL (Ni et al., 2020), EBIL (Liu et al., 2021), IQ-Learn (Garg et al., 2021) and GAIL (Ho & Ermon, 2016). Through these experiments, we examine two main scenarios: $(a)$ VLB-IRL's ability to learn an optimal reward function as compared to the baselines when noise-free optimal expert trajectories are available. $(b)$ VLB-IRL's capacity to generalize and learn a robust reward function as compared to the baselines, when noisy expert trajectories are provided.

### 4.1 MUJOCO DOMAINS

As part of our generalizability analysis, we consider both continuous and discrete control environments from the Mujoco library and use twin-delayed DDPG (TD3) (Fujimoto et al., 2018) and proximal policy optimization (PPO) (Schulman et al., 2017) for continuous and discrete domains respectively [2]. We readily obtain the expertly trained RL models from the popular RL library RL-Baselines3 Zoo (Raffin, 2020) and generate several expert trajectories employing them. Using the generated expert trajectories, each method was trained with 5 random seeds and for each seed, the model with the highest return was saved. At the end of the training, we evaluate these models on 50 test episodes. We use MLP for the reward function approximation. The hyperparameters and other relevant details needed for reproducibility are provided in the Appendix C. and our code is openly available on GitHub[3].

**Performance on optimal trajectories**   For our first set of experiments, we directly use the optimal expert trajectories obtained from the aforementioned pretrained models. Table 1 shows the summary of results obtained and the statistically best average reward accrued per domain is highlighted in bold. Our method VLB-IRL improves upon the baselines indicating that VLB-IRL's learned reward function successfully explains the expert's policy.

---

[2]The same RL algorithms are used for expert trajectory generation.
[3]Double blind review note: we will share our code once our paper gets accepted

Table 1: Performance when optimal trajectories are provided. The average return and standard deviation of the return are reported. The bold method statistically (t-test using a significance level of 0.01) outperforms other methods.

| Algorithm | LunarLander | Hopper | Walker2d | HalfCheetah | Ant |
|---|---|---|---|---|---|
| Random | $-195.92 \pm 97.17$ | $21.94 \pm 24.36$ | $1.16 \pm 6.26$ | $-282.06 \pm 83.09$ | $-59.50 \pm 103.75$ |
| GAIL | $255.58 \pm 8.51$ | $3518.80 \pm 54.12$ | $4529.11 \pm 102.53$ | $9551.28 \pm 96.01$ | $5321.00 \pm 26.81$ |
| AIRL | $239.11 \pm 55.03$ | $2611.74 \pm 890.14$ | $2524.16 \pm 824.00$ | $5342.70 \pm 1291.19$ | $4360.28 \pm 139.10$ |
| $f$-IRL | - | $3458.61 \pm 90.05$ | $4595.97 \pm 144.92$ | $9618.82 \pm 90.21$ | $4037.52 \pm 721.58$ |
| IQ-Learn | - | $3523.88 \pm 14.83$ | $4719.93 \pm 35.33$ | $9592.53 \pm 64.74$ | $5072.59 \pm 79.30$ |
| VLB-IRL(ours) | $\mathbf{267.99 \pm 10.85}$ | $\mathbf{3588.41 \pm 6.22}$ | $\mathbf{4779.76 \pm 32.25}$ | $\mathbf{9677.64 \pm 76.64}$ | $\mathbf{5422.10 \pm 83.02}$ |
| Expert | $266.67 \pm 12.73$ | $3606.22 \pm 4.06$ | $4835.76 \pm 52.30$ | $9714.77 \pm 154.35$ | $5547.41 \pm 1376.67$ |

**Learning performance** We find that VLB-IRL continues to perform well in a limited data regime. The empirical result demonstrates that VLB-IRL exhibits fairly good sample efficiency. With increasing amount of data, the standard deviation of the return from the learned policy becomes lower as shown in Figure 2, indicating better stability in the learned policy, which is a direct result of a better reward function. To further analyze the training progress of VLB-IRL, we use Inverse Learning Error (ILE) from Arora & Doshi (2021a), as $||V^{\pi_E} - V^{\pi_L}||_p$ where $\pi_E$ is the expert policy and $\pi_L$ is the learned policy. Here we use the $L2$ norm and the result is presented in Figure 2. The decreasing ILE shows that both the learned reward function is getting closer to the true reward function and the learned policy is more similar to the expert policy.

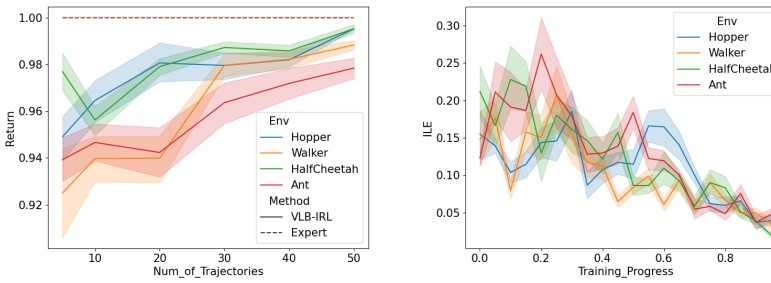

Figure 2: **Left**: Performance on different number of expert trajectories. The return has been normalized to $[0, 1]$. **Right**: ILE during the training. The training progress has been normalized to $[0, 1]$ due to different training steps for different environments. 0 represents the beginning of the training and 1 represents the end of the training.

**Performance on noisy trajectories** In order to further test VLB-IRL's generalizability, in our second set of experiments, we provide noisy trajectories and examine if the learned reward function is robust enough to learn the noisy expert's preferences and perhaps outperform them. The noisy trajectories are generated by using A2C for discrete environments and PPO for continuous environments in the library RL-Baselines3 Zoo. Compared to PPO for discrete environments and TD3 for continuous environments, they are less optimal and contain noisy actions in the trajectories. Compare the expert performance in Table 1 and Table 2 for detail. The results presented in Table 2 show that VLB-IRL's accrues comparable or higher reward on average as compared to the noisy expert.

For Hopper, HalfCheetah, and Ant environments, the VLB-IRL's learned policy outperforms the noisy expert by 15.6%, 14.4%, and 11.7% respectively. However, previous IRL techniques fail to generalize well when noisy trajectories are provided and we suspect that they fall short due to their optimization objective and divergence metric listed in Table 4.

## 4.2 REALISTIC SCENARIO

To further examine the performance of VLB-IRL, we use Assistive Gym (Erickson et al., 2020), a more realistic, robot-centered, and goal-based environment. For our experiments, we use environments with an active robot and a static human, such as FeedingSaywer, BedBathingSawyer, and ScratchItchSawyer. A description of the environments can be found in Figure.3. We use SAC to

Table 2: Performance when noisy trajectories are provided. The average return and standard deviation of the return are reported. The bold method statistically (t-test using a significance level of 0.01) outperforms other methods.

| Algorithm | LunarLander | Hopper | Walker2d | HalfCheetah | Ant |
|---|---|---|---|---|---|
| Random | $-195.92 \pm 97.17$ | $21.94 \pm 24.36$ | $1.16 \pm 6.26$ | $-282.06 \pm 83.09$ | $-59.50 \pm 103.75$ |
| GAIL | $238.11 \pm 18.10$ | $2644.29 \pm 412.57$ | $3489.23 \pm 211.55$ | $2682.65 \pm 366.92$ | $3884.23 \pm 938.26$ |
| $f$-IRL | - | $2371.61 \pm 236.14$ | $3603.85 \pm 164.74$ | $2980.33 \pm 124.99$ | $4140.15 \pm 508.96$ |
| EBIL | $\mathbf{239.47 \pm 54.81}$ | $2073.39 \pm 89.69$ | $3295.84 \pm 52.73$ | $2499.83 \pm 496.12$ | $1520.84 \pm 575.88$ |
| IQ-Learn | - | $2617.43 \pm 13.82$ | $3612.85 \pm 132.51$ | $3201.72 \pm 132.54$ | $4738.80 \pm 151.51$ |
| VLB-IRL(ours) | $237.61 \pm 7.66$ | $\mathbf{2786.22 \pm 13.57}$ | $\mathbf{3694.82 \pm 117.15}$ | $\mathbf{3375.81 \pm 109.29}$ | $\mathbf{5125.00 \pm 121.72}$ |
| Noisy Expert | $240.60 \pm 45.68$ | $2409.77 \pm 9.80$ | $3873.86 \pm 165.73$ | $2948.79 \pm 383.29$ | $4588.19 \pm 1258.34$ |

train an expert policy and use the trained expert policy to generate 50 expert episodes. The detailed hyperparameter can be found in Appendix C.

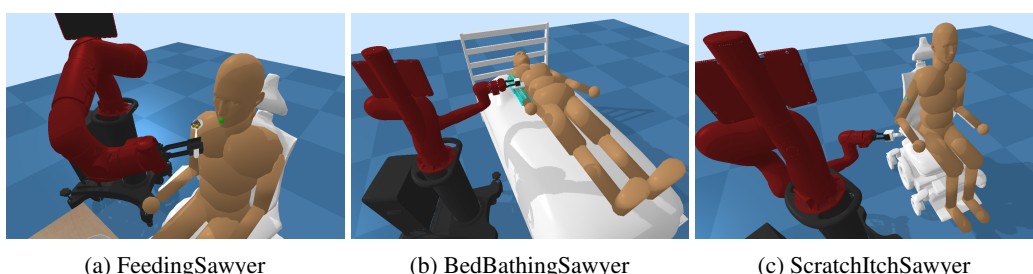

(a) FeedingSawyer       (b) BedBathingSawyer       (c) ScratchItchSawyer

Figure 3: Assistive gym environment. (a) A Sawyer robot holds a spoon with small spheres representing food on the spoon and must bring this food to a person's mouth without spilling it. (b) A person lies on a bed in a random resting position while a Sawyer robot must use a washcloth tool to clean off a person's right arm. (c) A Sawyer robot holds a small scratching tool and must reach towards a random target scratching location along a person's right arm.

The challenge in Assistive Gym compared to Mujoco benchmark is that it is a goal-based environment and the agent only receives a positive reward after the completion of the goal, unlike Mujoco where the agent can receive positive rewards at every timestep. This goal-based environment will generate much more noisy expert trajectories. Table 3 shows that VLB-IRL outperforms all other baselines and VLB-IRL is the only method that can successfully achieve the goal.

Table 3: Performance on Assistive Gym environments. The bold method statistically (t-test using a significance level of 0.01) outperforms other methods.

| Algorithm | FeedingSawyer | BedBathingSawyer | ScratchItchSawyer |
|---|---|---|---|
| Random | $-106.39 \pm 9.21$ | $-21.42 \pm 3.26$ | $-31.74 \pm -5.50$ |
| GAIL | $-50.68 \pm 76.33$ | $-6.63 \pm 14.38$ | $-23.75 \pm 6.48$ |
| AIRL | $-76.13 \pm 15.58$ | $-15.22 \pm 6.95$ | $-27.85 \pm 8.40$ |
| $f$-IRL | $-65.38 \pm 19.64$ | $-9.69 \pm 19.82$ | $-25.91 \pm 5.08$ |
| IQ-Learn | $-30.33 \pm 49.39$ | $-2.34 \pm 11.93$ | $-15.32 \pm 9.55$ |
| VLB-IRL(ours) | $\mathbf{88.11 \pm 52.95}$ | $\mathbf{10.86 \pm 13.94}$ | $\mathbf{11.94 \pm 24.44}$ |
| Expert | $117.74 \pm 30.42$ | $67.60 \pm 37.22$ | $61.64 \pm 29.16$ |

## 5 RELATED WORK

IRL was introduced more than two decades ago and Abbeel & Ng (2004)'s Apprenticeship Learning, which sought to learn the reward function with the maximum margin, provided significant early impetus to IRL's development. Subsequently, Bayesian methods for IRL, which viewed the trajectories as observations, were introduced Ramachandran & Amir (2007). Subsequently, Choi & Kim (2011) searches for the maximum-a-posteriori reward function instead of settling for the posterior mean. As it is hard to integrate over the entire reward space, the posterior mean may not be an ideal proposal

for the reward inference. However, Bayesian methods are now known to be severely impacted by the issue of being unable to scale to large domains. Toward this challenge, recently Chan & van der Schaar (2021) scales Bayesian IRL to learn in the context of complex state spaces and mitigates reward uncertainty by using a variational approximation of the posterior distribution over reward and can be executed entirely offline.

Fu et al. (2018a) presents adversarial IRL (AIRL), a practical and scalable IRL algorithm based on an adversarial reward learning formulation, which has received significant attention and is one of our baseline techniques in this paper. More recently, Ghasemipour et al. (2019) demonstrated that the objective in AIRL is equivalent to minimizing the reverse KL divergence between the joint state-action distribution induced by the learned policy and that of the expert. However, this has been disputed by Ni et al. (2020), which claims that AIRL is not optimizing reverse KL divergence in the state-action marginal. Ni et al. (2020) also introduces $f$-IRL that optimizes the $f$-divergence measure, and is one of our baseline methods as well.

Although VLB-IRL and AIRL can be identified as adversarial method, VLB-IRL and AIRL come from different frameworks. AIRL comes from generative adversarial network guided cost learning (GAN-GCL) and the discriminator is interpreted as $D_{\theta,\phi} = \frac{\exp(f_\theta)}{\exp(f_\theta) + \pi_\phi}$. By training $D_{\theta,\phi}$ via binary logistic regression and placing a special structure, i.e. advantage function, on the discriminator, the reward can be recovered. VLB-IRL comes from the graphical model and the graphical model naturally hold the conditional independence, leading to two proposed interpretation of optimality, i.e. $q(\mathcal{O}_t|r_t)$ and $p(\mathcal{O}_t|s_t, a_t)$. By optimizing the reverse KL divergence in ELBO of log-likelihood of trajectory $p(\tau)$, i.e. $KL(q(\mathcal{O}_t|r_t) \parallel p(\mathcal{O}_t|s_t, a_t))$, the reward can be recovered. In practice, VLB-IRL is more flexible. AIRL only accepts stochastic policy, while VLB-IRL is compatible with both stochastic and deterministic policy.

More recently, IQ-Learn has been proposed by Garg et al. (2021). IQ-Learn is a method for dynamics-aware IL that avoids adversarial training by learning a single Q-function, implicitly representing both reward and policy. However, it is not an IRL method. It sacrifices reward estimation accuracy by indirectly recovering rewards through a soft Q-function approximator, which relies heavily on dynamic environmental factors and doesn't strictly adhere to the soft-Bellman equation.

Table 4: Relevant IRL and imitation learning methods.

| Algorithm | Optimization Space | Optimization Objective | Divergence measure | Type |
|---|---|---|---|---|
| GAIL | $s, a$ | $\rho(s, a)$ | Jensen-Shannon | IL |
| AIRL | $\tau$ | $\rho(s, a)$ | Forward KL | IRL |
| $f$-IRL | $s$ | $\rho(s)$ | $f$-divergence | IRL |
| EBIL | $\tau$ | $\rho(s, a)$ | Reverse KL | IRL |
| IQ-Learn | $s, a$ | $\rho(s, a)$ | $f$-divergence | IL |
| VLB-IRL(ours) | $s, a$ | $\mathcal{O}(s, a)$ | Reverse KL | IRL |

## 6 CONCLUSION

In this paper, we derive a novel variational lower bound on the log-likelihood of the trajectories for IRL and present an algorithm that maximizes this lower bound. By posting IRL through the framework of a probabilistic graphical model with an optimality node, the optimization dual problem becomes minimizing the reverse KL divergence between an approximate distribution of optimality given the reward function and the true distribution of optimality given the trajectories. In particular, it models IRL as an inference problem within the framework of PGMs. Our experiments demonstrate that VLB-IRL has better generalization in the presence of noisy expert trajectories compared to popular baselines. In addition, when (optimal) expert trajectories are available, VLB-IRL maintains a performance advantage compared to existing algorithms. The formulation of the graphical model opens up the possibility for a wide variety of new IRL techniques that can interpret the framework in various ways. Extending VLB-IRL to multi-agent environments and extending the interpretation of the optimality node for multi-agent learning is an intriguing direction.

A limitation of the VLB-IRL is that the tightness of its lower bound to the true log-likelihood is yet to be established, and VLB-IRL is not guaranteed to converge to the optimum, which is a limitation shared by many other adversarial IRL algorithms as well. Another common limitation of the adversarial algorithm is the unstable training progress.

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

## A    VARIATIONAL LOWER BOUND

$$\log p(\tau) = \mathbb{E}_q[\log p(\tau)]$$
$$= \mathbb{E}_q\left[\log \frac{p(\tau, \mathcal{O})]}{p(\mathcal{O}|\tau)}\right]$$
$$= \mathbb{E}_q\left[\log \frac{p(\tau, \mathcal{O})}{p(\mathcal{O}|\tau)} \frac{q(\mathcal{O}|r)}{q(\mathcal{O}|r)}\right]$$
$$= \mathbb{E}_q\left[\log \frac{p(\tau, \mathcal{O})}{q(\mathcal{O}|r)} \frac{q(\mathcal{O}|r)}{p(\mathcal{O}|\tau)}\right]$$
$$= \mathbb{E}_q\left[\log \frac{p(\tau, \mathcal{O})}{q(\mathcal{O}|r)}\right] + \mathbb{E}_q\left[\log \frac{q(\mathcal{O}|r)}{p(\mathcal{O}|\tau)}\right].$$

The first term on the RHS is exactly our ELBO and the second term on the RHS is the gap between ELBO and the true log-likelihood of expert trajectories.

$$\text{ELBO} = \mathbb{E}_q\left[\log \frac{p(\tau, \mathcal{O})}{q(\mathcal{O}|r)}\right]$$
$$= \mathbb{E}_q\left[\log \int_r p(\tau, \mathcal{O}, r) - \log q(\mathcal{O}|r)\right]$$
$$= \log p(s_1) + \sum_t \mathbb{E}_q\left[\log \int_{r_t} p(\mathcal{O}_t|r_t)p(r_t|s_t, a_t)p(s_{t+1}|s_t, a_t)p(a_t) - \log q(\mathcal{O}_t|r_t)\right]$$
$$= \log p(s_1) + \sum_t \mathbb{E}_q\left[\log p(\mathcal{O}_t|s_t, a_t) + \log p(s_{t+1}|s_t, a_t) + \log p(a_t) - \log q(\mathcal{O}_t|r_t)\right]$$
$$= \log p(s_1) + \sum_t [-\text{KL}(q(\mathcal{O}_t|r_t) \parallel p(\mathcal{O}_t|s_t, a_t)) + \log p(s_{t+1}|s_t, a_t) + \log p(a_t)].$$

## B    PROOF

### B.1    PROOF FOR LEMMA 1

*Proof.* Let $x_0 := \mathbb{E}[X]$ and we have the following Taylor expansion:

$$f(x) = f(x_0) + f'(x_0)(x - x_0) + \frac{f''(\xi(x))}{2}(x - x_0)^2 \tag{12}$$

where $\xi(x) \in (x_0, x)$. Replacing $x$ with the random variable $X$ and taking the expectation results in,

$$\mathbb{E}[f(x)] = f(x_0) + f'(x_0)\mathbb{E}[x - x_0] + \mathbb{E}\left[\frac{f''(\xi(x))}{2}(x - x_0)^2\right] = f(x_0) + \mathbb{E}\left[\frac{f''(\xi(x))}{2}(x - x_0)^2\right].$$

If function $f$ has a bounded second derivative, $|f''(x)| < M$ for all $x$, then we have

$$|\mathbb{E}[f(x)] - f(x_0)| = |\mathbb{E}[f(x)] - f(\mathbb{E}[x])| \le M\mathbb{E}[(X - x_0)^2] = M\text{Var}(X).$$

$\square$

### B.2    PROOF FOR THEOREM 2

*Proof.* As $\mathcal{O}_t$ is a binary variable, $p(\mathcal{O}_t|s_t, a_t)$ and $q(\mathcal{O}_t \mid \mathbb{E}[r_t])$ are Bernoulli distributions,

$$q(\mathcal{O}_t = 0 \mid \mathbb{E}[r_t]) = 1 - q(\mathcal{O}_t = 1 \mid \mathbb{E}[r_t])$$
$$p(\mathcal{O}_t = 0|s_t, a_t) = 1 - p(\mathcal{O}_t = 1|s_t, a_t)$$

We prove that $q(\mathcal{O}_t = 1 \mid \mathbb{E}[r_t])$ is a valid candidate for approximating $p(\mathcal{O}_t = 1|s_t, a_t)$.

$$p(\mathcal{O}_t = 1|s_t, a_t) = \int_{r_t} p(\mathcal{O}_t = 1|s_t, a_t, r_t)\, p(r_t|s_t, a_t) = \mathbb{E}_{r_t}[p(\mathcal{O}_t = 1|r_t)] \quad \text{(using Eq. 2)} \tag{13}$$

Let $f(r_t) := p(\mathcal{O}_t = 1|r_t)$, Eq. 13 now becomes

$$p(\mathcal{O}_t = 1|s_t, a_t) = \mathbb{E}_{r_t}[f(r_t)] \tag{14}$$

And

$$f(\mathbb{E}[r_t]) := p(\mathcal{O}_t = 1 \mid \mathbb{E}[r_t]) \tag{15}$$
$$= q(\mathcal{O}_t = 1 \mid \mathbb{E}[r_t]) \tag{16}$$

Given the condition $|p''(\mathcal{O}_t = 1|r_t)| < M$, Eq. 14 and Eq. 16, applying Lemma 1, we get

$$|\mathbb{E}[f(r_t)] - f(\mathbb{E}[r_t])| = |p(\mathcal{O}_t = 1|s_t, a_t) - q(\mathcal{O}_t = 1 \mid \mathbb{E}[r_t])| \leq M\text{Var}(r_t) \tag{17}$$

$\square$

### B.3 PROOF FOR THEOREM 3

*Proof.* If $p(\mathcal{O}_t|s_t, a_t)$ is maximal, from Lemma 2, we obtain that

$$a_t^* = \arg\max_{a_t} p(\mathcal{O}_t = 1|s_t, a_t) \tag{18}$$

In addition, if $q_\phi(\mathcal{O}_t|r_t)$ is identical to $p_\theta(\mathcal{O}_t|s_t, a_t)$,

$$\begin{aligned}
a_t^* &= \arg\max_{a_t} p(\mathcal{O}_t = 1|s_t, a_t) \\
&= \arg\max_{a_t} q_\phi(\mathcal{O}_t = 1|r_t) \quad \text{(given the assumption above)} \\
&= \arg\max_{a_t} \sigma(A_\phi(s_t, a_t)).
\end{aligned} \tag{19}$$

Here, $A_\phi(s_t, a_t^*)$ parameterized by $r_\phi(s_t, a_t)$ has the highest value given $s_t$ for any $a_t$, which best explains the expert policy $\pi^E$. $\square$

## C IMPLEMENTATION AND HYPERPARAMETER

### C.1 IMPLEMENTATION

In this section, we discuss several implementation details for making training stable and getting a better reward function and learner policy.

**Iterative updates.** Many algorithms for IRL exhibit a nested structure, involving an inner loop and an outer loop. The inner loop focuses on finding the optimal policy given parameterized rewards. However, this step is costly and we can avoid this by applying the iterative updates. We borrow the idea from Finn et al. (2016), updating the reward function and learning policy in parallel without finding the optimal policy given parameterized rewards.

**Learner Policy Rollouts Buffer.** There are two reasons to use the learner policy rollouts buffer. First, recall that in Section 2.3, we get the approximation of the distribution of optimality given trajectory, $p(\mathcal{O}_t|s_t, a_t)$ by using a classifier $C_\theta$ classifying whether a trajectory comes from an expert or learner. In supervised learning with deep neural networks, it is common practice to compute gradient estimates based on a minibatch of samples rather than individual samples. It is essential to ensure that the samples are independently and identically distributed (i.i.d) from a fixed distribution. However, if we sample minibatches from the most recent learner policy, the i.i.d assumption cannot be held anymore because they are correlated. This is similar to the scenario described in Mnih et al. (2013). By applying the learner policy rollouts buffer, we can alleviate the problems of correlated data and non-stationary distributions. Second, to avoid catastrophic forgetting, which impedes the training process heavily, the simplest solution is to store past experiences. By including previous experiences in the training set, the neural network benefits from a diverse range of samples, leading to a reduced risk of catastrophic forgetting.

## C.2 HYPERPARAMETER AND OTHER IMPLEMENTATION DETAILS

**Environment:** We use the `LunarLander-v2`, `Hopper-v3`, `Ant-v3`, `HalfCheetah-v3`, `Walker2d-v3` from OpenAI Gym. We use `FeedingSawyer-v1`, `BedBathingSawyer-v1`, `ScratchItchSawyer-v1` from Assistive Gym.

**Expert Policy:** For OpenAI Gym environments, we readily obtain the expertly trained RL models from the popular RL library RL-Baselines3 Zoo (Raffin, 2020). We use TD3 and PPO for continuous and discrete action environments for optimal trajectories collection respectively. We use PPO and A2C for continuous and discrete action environments for sub-optimal trajectories collection respectively. For Assistive Gym environments, we train the expert policy using SAC from Stable-Baselines3. We use policy network as $[512, 512]$ and action noise with $\mathcal{N}(0, 0.2)$. The remaining hyperparameter remains the default value.

**Training Details:** We train 6 algorithms namely VLB-IRL, IQ-Learn, GAIL, AIRL, $f$-IRL, and EBIL to recover the expert reward function and imitate the expert behavior using the given expert trajectories.

For OpenAI Gym environments, we train VLB-IRL using Algorithm 1. We use TD3 and PPO for continuous and discrete action environments from the popular RL library Stable-Baselines3 respectively. The hyperparameter for learner policy is listed in Table 5 and Table 7 and the unmentioned hyperparameters are the default value in the Stable-Baselines3 package. The hyperparameters for learner policy are borrowed from the RL-Baselines3 Zoo, which includes a collection of hyperparameters for various kinds of algorithms and environments. The hyperparameter for the reward function network and the classifier are listed in Table 8 and Table 10 respectively.

For Assistive Gym environments, we train VLB-IRL using Algorithm 1. We use SAC from the popular RL library Stable-Baselines3. The hyperparameter for learner policy is listed in Table 6

For GAIL and AIRL, we use the Python package `Imitation`, which provides clean implementations of imitation and reward learning algorithms. For IQ-Learn, EBIL, and $f$-IRL, we refer to their authors' official implementation. We use the same hyperparameter setting for learner policy and reward function network. as VLB-IRL except for AIRL. AIRL only accepts stochastic policy, which is not true for TD3. Therefore, we use SAC as the underlying RL algorithm. The hyperparameter for SAC is borrowed from Ni et al. (2020).

Table 5: Hyperparameter setting for learner policy for continuous action OpenAI Gym environments.

| TD3 | Hopper | Ant | HalfCheetah | Walker2d |
|---|---|---|---|---|
| Learning rate | $1e^{-3}$ | $1e^{-3}$ | $1e^{-3}$ | $1e^{-3}$ |
| Gamma | 0.99 | 0.98 | 0.98 | 0.98 |
| Batch size | 256 | 256 | 256 | 256 |
| Gradient steps | 1000 | 1000 | 1000 | 1000 |
| Net arch | [400, 300] | [400, 300] | [400, 300] | [400, 300] |
| Buffer size | 2000000 | 200000 | 200000 | 200000 |
| Action noise | $\mathcal{N}(0, 0.2)$ | $\mathcal{N}(0, 0.2)$ | $\mathcal{N}(0, 0.2)$ | $\mathcal{N}(0, 0.2)$ |

Table 6: Hyperparameter setting for learner policy for Assistive Gym environments.

| SAC | FeedingSawyer | BedBathingSawyer | ScratchItchSawyer |
|---|---|---|---|
| Learning rate | $3e^{-4}$ | $3e^{-4}$ | $3e^{-4}$ |
| Gamma | 0.97 | 0.97 | 0.97 |
| Batch size | 256 | 256 | 256 |
| Net arch | [512, 512] | [512, 512] | [512, 512] |
| Action noise | $\mathcal{N}(0, 0.2)$ | $\mathcal{N}(0, 0.2)$ | $\mathcal{N}(0, 0.2)$ |

Table 7: Hyperparameter setting for learner policy for discrete action OpenAI Gym environments.

| PPO | LunarLander |
|---|---|
| Learning rate | $1e^{-3}$ |
| Gamma | 0.99 |
| Batch size | 100 |
| Entropy coefficient | $1e^{-2}$ |

Table 8: Hyperparameter setting for reward function network in OpenAI Gym environments.

| MLP | LunarLander | Hopper | Ant | HalfCheetah | Walker2d |
|---|---|---|---|---|---|
| Learning rate | $1e^{-3}$ | $1e^{-3}$ | $1e^{-3}$ | $1e^{-3}$ | $1e^{-3}$ |
| Net arch | [16, 16] | [16, 16] | [32, 32] | [32, 32] | [32, 32] |
| Optimizer | Adam | Adam | Adam | Adam | Adam |
| Batch size | 64 | 256 | 256 | 256 | 256 |
| Activation | LeakyReLU | LeakyReLU | LeakyReLU | LeakyReLU | LeakyReLU |

Table 9: Hyperparameter setting for reward function network in Assistive Gym environments.

| MLP | FeedingSawyer | BedBathingSawyer | ScratchItchSawyer |
|---|---|---|---|
| Learning rate | $1e^{-3}$ | $1e^{-3}$ | $1e^{-3}$ |
| Net arch | [64, 64] | [64, 64] | [64, 64] |
| Optimizer | Adam | Adam | Adam |
| Batch size | 256 | 256 | 256 |
| Activation | LeakyReLU | LeakyReLU | LeakyReLU |

Table 10: Hyperparameter setting for the classifier in OpenAI Gym environments.

| MLP | LunarLander | Hopper | Ant | HalfCheetah | Walker2d |
|---|---|---|---|---|---|
| Learning rate | $1e^{-3}$ | $1e^{-3}$ | $1e^{-3}$ | $1e^{-3}$ | $1e^{-3}$ |
| Net arch | [16, 16] | [16, 16] | [32, 32] | [32, 32] | [32, 32] |
| Optimizer | Adam | Adam | Adam | Adam | Adam |
| Batch size | 64 | 256 | 256 | 256 | 256 |
| Activation | LeakyReLU | LeakyReLU | LeakyReLU | LeakyReLU | LeakyReLU |

Table 11: Hyperparameter setting for the classifier in Assistive Gym environments.

| MLP | FeedingSawyer | BedBathingSawyer | ScratchItchSawyer |
|---|---|---|---|
| Learning rate | $1e^{-3}$ | $1e^{-3}$ | $1e^{-3}$ |
| Net arch | [64, 64] | [64, 64] | [64, 64] |
| Optimizer | Adam | Adam | Adam |
| Batch size | 256 | 256 | 256 |
| Activation | LeakyReLU | LeakyReLU | LeakyReLU |

