# OpenReview forum: "A Novel Variational Lower Bound For Inverse Reinforcement Learning"
_ICLR.cc/2024/Conference — Submitted to ICLR 2024_

### Official Review · Reviewer_Ed3u · 2023-10-25

**Soundness:** 1 poor
**Presentation:** 2 fair
**Contribution:** 1 poor
**Rating:** 1
**Confidence:** 3

**Summary:**

Inspired by Levine (2018), the paper proposes a probabilistic graphical model for IRL by introducing reward and optimality nodes. Then, it proposes a novel variational lower bound which leads to a new IRL solution. The resulting IRL method is shown to outperform state-of-the-art IRL algorithms in several discrete and continuous environments from the Mujoco library.

**Strengths:**

The use of variational inference and the optimality node the graphical model of IRL seems to be novel. The empirical performance of the method is promising.

**Weaknesses:**

The probabilistic graphical model is not very convincing. The derivation of ELBO in variational inference (VI) does not seem to follow the standard VI derivation. Hence, it raises a major concern whether the technical approach is correct.

1. In RL and IRL, there is only 1 reward function, so $r_t$ and $r_{t'}$ should be related through the reward function parameters. Hence, the graphical model may be clearer if we include the reward function parameter node. Right now, it is unclear to me why we have the conditional independence of $\mathcal{O}_t$ from $a_t$ and $s_t$ given $r_t$. In Levine (2018), $\mathcal{O}_t$ is the optimality of the state-action pair given the reward function.

2. In the standard VI framework, it is often that we cannot directly minimize the KL[q(Z)||p(Z|X)] (supposed that we are interested in finding the posterior distribution of Z given the observation X). That is why we need to construct an ELBO that does not involve this KL term. Surprisingly, the ELBO formulation in equation (4) directly involves the term KL[q(Z)||p(Z|X)], i.e., the term KL[q(Ot|rt) || p(Ot|st,at)] in equation (4).

3. We note that $\int_{\mathcal{O}_t} p(\mathcal{O}_t|s_t,a_t) = 1$ for any distribution $p(\mathcal{O}_t|s_t,a_t)$. Furthermore, the authors claim that all terms except the KL terms are constants w.r.t.~$r_t$ (at the end of page 3). Then, any distribution $p(\mathcal{O}_t|s_t,a_t)$ satisfies the derivation in the beginning of Section 2.2. Hence, it does not make sense to find $r_t$ by minimizing the distance between $q(\mathcal{O}_t|r_t)$ and $p(\mathcal{O}_t|s_t,a_t)$.

**Questions:**

Please address the above weaknesses.

---

> ### Author Response · Authors · 2023-11-20
>
> Thank you for your response regarding our paper review. Below, you'll find our replies to your comments and feedback.
>
> 1. Reward and optimality.
>
> There is only 1 reward function which is expressed in the node $R$ and it is not conditioned on time $t$. In the revised paper, we will separate the $R$ as a single node connecting to all $r_t$ nodes. In Levine (2018), $\mathcal{O}\_t$ is conditioned $r_t$. However, in IRL setting, $r_t$ is unknown. Therefore, we need to know $\mathcal{O}\_t$ given $s_t, a_t$, i.e. $p(\mathcal{O}\_t|s_t,a_t)$. Given $p(\mathcal{O}\_t|s_t, a_t)$, we need to find the reward function $R$ such that $q(\mathcal{O}\_t|r_t)$ is approximating $p(\mathcal{O}\_t|s_t, a_t)$. Thus, $R$ best explains the expert trajectories.
>
> 2. VI framework.
>
> Note that KL$(q(Z)||p(Z|X))$ is different from our KL divergence. Here is the derivation
> $$
> \begin{align*}
>     \log p(\tau) &= \mathbb{E}_q[\log p(\tau)]\\\\
>     &= \mathbb{E}_q\left[\log \frac{p(\tau,\mathcal{O})]}{p(\mathcal{O}|\tau)}\right] \\\\
>     &= \mathbb{E}_q\left[\log\frac{p(\tau,\mathcal{O})}{p(\mathcal{O}|\tau)}\frac{q(\mathcal{O})}{q(\mathcal{O})}\right]\\\\
>     &= \mathbb{E}_q\left[\log \frac{p(\tau,\mathcal{O})}{q(\mathcal{O})}\frac{q(\mathcal{O})}{p(\mathcal{O}|\tau)}\right]\\\\
>     &= \mathbb{E}_q\left[\log\frac{p(\tau,\mathcal{O})}{q(\mathcal{O})}\right]+\mathbb{E}_q\left[\log\frac{q(\mathcal{O})}{p(\mathcal{O}|\tau)}  \right].
>     \end{align*}
> $$
> The first term on the RHS is exactly our ELBO and the second term on the RHS is the gap between ELBO and the true log-likelihood of expert trajectories. The second term is intractable because it involves calculating the marginal distribution of evidence, $p(\tau)$.
>
> 3. $p(\mathcal{O}\_t|s_t, a_t)$.
>
> The true distribution $p(\mathcal{O}\_t|s_t,a_t)$ is not an arbitrary distribution. It is determined by an unknown expert policy. In fact, the state-action pairs from the expert policy will have the highest probability from $p(\mathcal{O}|s_t, a_t)$ and all other state-action pairs will have relatively low probability. In our paper, we use classifier $C_\theta$ to approximate such a true distribution. Hence, minimizing the distance between $q$ and $p$ holds the necessity.

---

> > ### Comment · Reviewer_Ed3u · 2023-11-20
> >
> > The notations remain confusing to me. Could the authors ensure consistency in the notation used throughout the paper and the response?
> >
> > 1. Is $\mathcal{R}$ considered a random variable or a random function? What is the relationship between $\mathcal{R}$ and $r_t$? I initially assumed that the reward $r_t$ at time $t$ should be computed based on the reward function $\mathcal{R}$ and the state-action pair $s_t$ and $a_t$. If this assumption is correct, given $r_t$, $\mathcal{O}_t$ is independent of $\mathcal{R}$. In that case, how can we determine the reward function $R$ such that $q(\mathcal{O}_t|r_t)$ (does not depend on $\mathcal{R}$) approximates $p(\mathcal{O}_t|s_t,a_t)$?
> >
> > In Section 2.3, what is $C_\theta$?, and $\mathcal{R}(s_t,a_t) = \mathcal{N}(\mu,\sigma^2)$ is defined. Why does $\\mathcal{R}(s\_t, a\_t)$ not depend on $s\_t$ and $a\_t?
> >
> > It is surprising that $\\mathcal{R}(s\_t, a\_t)$ follows the same distribution for all $s_t$ and $a_t$.
> >
> > What does it mean when $r_t \sim \mathcal{R}(s_t,a_t)$ in Section 2.3? Since both are random variables, should it not be expressed as $r_t = \mathcal{R}(s_t,a_t)$?
> >
> > 2. Could the authors ensure consistency in notations with those presented in the paper and provide a comprehensive derivation of the ELBO from the RHS mentioned above?
> >
> > If we assume that the posterior $p(\mathcal{O}|\tau)$ is the one approximated by the VI framework, then the expansion of $p(\tau,\mathcal{O})$ should be $p(\tau|\mathcal{O}) p(\mathcal{O})$ (the first term being the likelihood and the second term being the prior). However, it seems the authors express $p(\tau,\mathcal{O}) = p(\mathcal{O}|\tau) p(\tau)$ which does not make sense to me. If we can compute the posterior $p(\mathcal{O}|\tau)$ directly, why do we need to use VI? This is also related to the third response; if the authors already approximate the posterior in a closed-form expression with $C_\theta$, then why employ VI?
> >
> > 3. The response from the authors fails to address my question. As stated in my question, when examining the derivation at the start of Section 2.2 (the line ending with "using Eq. 2"), we can substitute any arbitrary distribution $p(\mathcal{O}_t|s_t,a_t)$ because $\\int\_{\\mathcal{O}\_t} p(\\mathcal{O}\_t|s\_t,a\_t) = 1$ holds true for any distribution $p(\mathcal{O}_t|s_t,a_t)$. Hence, it appears that utilizing VI is incorrect.

---

> > > ### Author Response · Authors · 2023-11-22
> > >
> > > We have updated our paper with a new graphical model, in which we separate out the reward function node and make it a deterministic node.
> > >
> > > 1. $R$ represents the reward function and it is now a deterministic node instead of a random node in the updated paper. There is only one reward function node $R$ in the graphical model. Every reward value $r_t$ comes from the same reward function $R$. And there is not any randomness for the reward function. However, the output of the reward function is a Gaussian distribution, denoted by $R(s_t, a_t)$. In conclusion, $R$ is the reward function. $R(s_t, a_t)$ is the reward value distribution given $s_t, a_t$ as input to the reward function $R$. $r_t$ is the sampled reward value from the reward distribution $R(s_t, a_t)$.
> > >
> > > "I initially assumed that the reward $r_t$ at time $t$ should be computed based on the reward function $R$ and the state-action pair $s_t$ and $a_t$." This is correct.
> > >
> > > "If this assumption is correct, given $r_t$, $\mathcal{O}\_t$ is independent of $R$." This is not correct. Reward function $R$ is not a random variable. The reason why there exists a $R$ node in the graphical model is to show that every reward value comes from the single reward function $R$.
> > >
> > > "how can we determine the reward function $R$ such that $q(\mathcal{O}\_t|r_t)$ approximates $p(\mathcal{O}\_t|s_t,a_t)$?" First of all, the reward function $R$ is no longer a random variable. Moreover, The reward value is sampled from the reward distribution $R(s_t, a_t)$ and the reward distribution is the output of the reward function $R$. Therefore, $q(\mathcal{O}\_t|r_t)$ is connected with the reward function $R$, and by backpropagation and reparameterization, we can update the parameter of the reward function.
> > >
> > > $C_\theta$ serves as a binary classifier wherein it accepts $s_t$ and $a_t$ as input variables and produces the Bernoulli distribution as an output, which denotes the probability associated with optimality.
> > >
> > > $R(s_t, a_t)$ depends on $s_t$ and $a_t$ since it is the output of the reward function given $s_t, a_t$. We are unable to locate the sentence in which we claim that $R(s_t, a_t)$ does not depend on $s_t$ and $a_t$.
> > >
> > > $R(s_t, a_t)$ does not follow the same distribution for all $s_t$ and $a_t$. Every state-action pair $s_t,a_t$ generates a different reward value distribution $R(s_t,a_t)$. In detail, the reward value distribution $R(s_t, a_t)$ is $\mathcal{N}(\mu(s_t,a_t),\sigma^2(s_t,a_t))$.
> > >
> > > "What does it mean when $r_t\sim R(s_t,a_t)$ in sec 2.3?" $r_t$ is the reward value sampled from the reward value distribution $R(s_t, a_t)$ using reparameterization. We cannot express it by $r_t=R(s_t,a_t)$.
> > >
> > > 2. We add the detailed derivation for the ELBO in Appendix A. We agree that our derivation for the ELBO does not follow the standard VI framework. However, this deviation takes advantage of the graphical model, which is the connection between $p(\mathcal{O}\_t|s_t,a_t)$ and $q(\mathcal{O}\_t|r_t)$, and makes $r_t$ explicitly incorporated into the ELBO, which has never been done before.
> > >
> > > "Why do we need to use VI?" Even if we have already approximated the posterior in a closed-form expression with $C_\theta$, we still do not know the connection between the optimality and the reward function, which is the goal of IRL. Employing VI helps us retrieve such a connection. Namely, we get a reward function after the approximation.
> > >
> > > 3. The definition of $p(\mathcal{O}\_t|s_t,a_t)$.
> > >
> > > For the derivation at the start of Sec 2.2. Without the IRL setting, it is correct that $p(\mathcal{O}\_t|s_t,a_t)$ can be substituted by any distribution. However, once the expert trajectories are given in the IRL setting, $p(\mathcal{O}\_t|s_t,a_t)$ is determined by the expert trajectories. There is only one $p(\mathcal{O}\_t|s_t,a_t)$ best explaining the expert trajectories. This is similar to the occupancy measure $\rho(s,a)$ in the prior works. We cannot claim that we can use arbitrary occupancy measure $\rho(s,a)$ to represent the expert trajectories. Both occupancy measure $\rho(s,a)$ and $p(\mathcal{O}\_t|s_t,a_t)$ are determined at the same time once the expert trajectories are given. However, we do not have access to $p(\mathcal{O}\_t|s_t,a_t)$, so we use the classifier $C_\theta$ to approximate the true distribution $p(\mathcal{O}\_t|s_t,a_t)$.

---

> > > > ### Comment · Reviewer_Ed3u · 2023-11-22
> > > >
> > > > After reading the authors' second response, my opinion remains that major revisions are needed to ensure a clear and accurate expression of its formulation and graphical model.
> > > >
> > > > ```
> > > > And there is not any randomness for the reward function. However, the output of the reward function is a Gaussian distribution $R(s_t,a_t)$
> > > > ```
> > > > VI is used to estimate the posterior distribution of a latent (random) variable given the data. In this paper, the reward function is the latent variable (or the parameters of $\mu$ and $\sigma^2$ if they are functions from $(s,a)$ to the mean and variance of the reward at $(s,a)$) but these latent variables are deterministic.
> > > > Therefore, it raises questions about the applicability of the VI framework in this context.
> > > > Moreover, it appears odd to me to place a distribution ($\mathcal{R}$) in a node of the graphical model. Saying a function's output is a distribution also seems odd. Usually, people represent the parameters, whether deterministic or random, of a distribution as nodes in the graphical model. However, this could due to my limited knowledge.
> > > >
> > > > ```
> > > > We are unable to locate the sentence where we claim that $R(s_t,a_t)$ does not depend on $s_t$ and $a_t$
> > > > ```
> > > > In the paragraph following equation (8), $\mathcal{R}(s_t,a_t) = \mathcal{N}(\mu,\sigma^2)$ and $r_t = \mu + \sigma \epsilon$, none of the RHS depends on $s_t$ nor $a_t$. Hence, the $\mathcal{R}(s_t,a_t)$ is the same for all $s_t$,$a_t$.
> > > >
> > > > The derivation of ELBO in Appendix A lacks sufficient detail and clarification, particularly concerning
> > > > - the connection $\mathcal{O}$ between $\mathcal{O}_t$,
> > > > - the relationship between $q(\mathcal{O}|r)$ and $q(\mathcal{O}_t|r_t)$,
> > > > - the transition from the second to the third line in the ELBO derivation where $\mathcal{O}_t$ is introduced.
> > > > - It is unclear about $p(a_t)$ and its consistency, because $p(a_t)$ was corrected to $p(a_t|s_t)$ in the last revision but reverted back to $p(a_t)$ in the current revision. It is very confusing.
> > > > - Why does $p(a_t|s_t)$ or $p(a_t)$ not depend on $r_t, as indicated by the partial derivative in equation 6?
> > > >
> > > > ```
> > > > we still do not know the connection between the optimality and the reward function
> > > > ```
> > > > In equation (8), $q(\mathcal{O}_t =1|r_t) \propto e^{r_t}$, so the paper enforces a connection and it is not a result of the VI framework.

---

### Official Review · Reviewer_PBAj · 2023-10-30

**Soundness:** 1 poor
**Presentation:** 2 fair
**Contribution:** 1 poor
**Rating:** 1
**Confidence:** 5

**Summary:**

The paper proposes a method for imitation learning that extends GAIL by introducing a second reward model that approximates the discriminator reward. This reward model also aims to unshape the GAIL reward based on the value function learnt by the RL algorithm.
The proposed method VB-IRL is evaluated  on MuJoCo and AssistiveGym environments, where we slightly outperforms baseline methods such as IQ-Learn.

**Strengths:**

I am not aware of prior work that used the RL value function for unshaping the GAIL reward. Also using a KL loss to approximate a second reward model seems to be novel.

**Weaknesses:**

1. Soundness
------------------
a) The derivations are based on several wrong assumptions, namely, that
- the expert actions do not depend on the state
- the probability of an action being optimal is conditionally independent of state and action given its immediate

These assumptions are not clearly communicated, but instead hidden in a graphical model, that is presented as a matter of fact.

b) There seems to be a mistake in Eq. 9, as it essentially states that $A(s,a)=\sigma(A(s,a))$. Furthermore, eq. 10 uses $\sigma(\exp(A(s,a))$. There also seems to be an expectation over $s_{t+1}$ missing, unless deterministic dynamics are implicitly assumed!?

c) Eq. 6 defines the "true distribution" $p(\mathcal{O}_t|s_t, a_t)$ as the discriminator reward. This is wrong, because the derivations use $p(\mathcal{O}_t|s_t, a_t)$ as the actual distribution of the optimality event given state and action (according to the graphical model). The discriminator reward is at best an approximation of this probability. Correspondingly, Theorem 2 is wrong, as it dot not bound the approximation between $q(\mathcal{O}_t|r_t)$ and the true distribution $p(\mathcal{O}_t|s_t, a_t)$. Furthermore, in the proof of Theorem 2, it is not clear why Eq. 15 holds.

2. Presentation
--------------------
a) $C_{\theta}(s,a)$ is not defined. The paper just states "To make this classification, we may simply use binary logistic regression, $C_{\theta}(s,a)$. $C_{\theta}(s,a)$ needs to be explicitly defined as a function of the discriminator logits (or the discriminator output if that is more convenient). If $C_{\theta}(s,a)$ directly corresponds to the discriminator output, this needs to be clearly stated.

b) It is not clear to me whether the reward network outputs a scalar, or a distribution (e.g. mean & std of a Gaussian). The paper states "To illustrate this, consider the simplistic case where the reward value distribution is a univariate Gaussian: [...]", but it doesn't explicitly state that a Gaussian distribution is used in the experiments.

3. Evaluation
-----------------
a) The paper claims that the results are significant according to a t-test, but it doesn't provide any further information on how the t-test was performed. It is not clear to me how the t-test was performed, because a t-test is usually used for comparing two groups. Were independent t-test performed for each combination of two groups? In general, I don't think that a t-test is appropriate in this case, not only because of non-Gaussian distributions, but in particular because the different populations can have significantly different variance.

b) The evaluations are performed by using the best policy among 5 different seeds. Such procedure may favor unstable methods. It would be better to compare the average performance among different seeds.

c) The paper stresses the performance of VB-IRL on noisy demonstrations and also evaluates the method in this setting. I think it would be fair to include methods that are targeted at this problem setting, e.g. [Sasaki, F., & Yamashina, 2020].

d) The paper does not show learning curves. These need to be added, at least to the appendix.

e) Ablations are missing with respect to $\lambda$ and also to evaluate the effect of unshaping the reward function with the value function.


References
---------------
Sasaki, F., & Yamashina, R. (2020, October). Behavioral cloning from noisy demonstrations. In International Conference on Learning Representations.

**Questions:**

How exactly are $C_{\theta}(s,a)$ and $r_t$ computed?

How is the t-test performed?

---

> ### Author Response · Authors · 2023-11-20
>
> Thank you for your response regarding our paper review. Below, you'll find our replies to your comments and feedback.
>
> 1. The assumptions.
>
> We have updated our paper, in which there is an edge between state and action, and corresponding modification in the derivation has been made. However, adding such an edge does not have any impact on our ELBO. What's more, the two assumptions are discussed in the original Levine's paper (Reinforcement Learning and Control as Probabilistic Inference: Tutorial and Review) (listed as "control as inference paper" in the following). The first assumption is addressed by forward and backward message passing in section 2.3 in "control as inference paper" and expressed by the following
>
> $$
>     \begin{align*}
>         \beta_t(s_t, a_t) &= p(\mathcal{O}\_{t:T}|s_t, a_t) \\\\
>         \beta_t(s_t) &= p(\mathcal{O}\_{t:T}|s_t)\\\\
>         p(a_t|s_t, \mathcal{O}\_{t:T}) &= \frac{\beta_t(s_t, a_t)}{\beta_t(s_t)}
>     \end{align*}
> $$
>
> The second assumption follows the setting in the eqn.3 in "control as inference paper"
> $$p(\mathcal{O}\_t=1|s_t, a_t) = \exp(r(s_t, a_t))$$
>
> 2. Advantage function.
>
> We try to express that the first term and third term in Eq.9 are equal, namely $q(\mathcal{O}\_t=1|r_t)=\sigma(A(s_t, a_t))$. Since this is an IRL setting, what we have is expert trajectories and we do not know the transition function of the environment. Therefore, the expectation over $s\_{t+1}$ has been omitted. However, this is not equal to admit that the environment is deterministic. This statement is consistent with the setting of the AIRL paper, where the discriminator tries to optimize the reward function by maximizing the advantage function.
>
> 3. Eq.6.
>
> Eq.6 is not the discriminator reward. $C_\theta$ is a binary classifier classifying whether the state-action pair comes from an expert or not and serves as the true distribution as we do not have access to the real true distribution $p(\mathcal{O}\_t|s_t, a_t)$. $C_\theta$ outputs a Bernoulli distribution. Indeed, $q(\mathcal{O}\_t|r_t)$ is at best an approximation of $C_\theta$. Therefore, if the variance of $r_t$ is zero, Theorem 2 states that the approximation error vanishes between $q(\mathcal{O}\_t|r_t)$ and $p(\mathcal{O}\_t|s_t,a_t)$, which means $q(\mathcal{O}|r_t)$ seamlessly approximates $p(\mathcal{O}\_t|s_t, a_t)$, leading to a learned reward best explaining the expert trajectories. Eq.15 and eq.14 are the same except for the notation. We are sorry for the misunderstanding.
>     $$f(\mathbb{E}[r_t]):=q(\mathcal{O}\_t=1|\mathbb{E}[r_t])$$.
>
> 4. Reward network.
>
> The reward network outputs a Gaussian distribution and such a Gaussian distribution is used in the experiments. Outputting a reward distribution improves the performance of our algorithms. The assumption on why it improves the performance is the randomness makes the learned reward function change smoothly since the model optimizes both the mean and variance. Updating the mean usually leads to a higher variance. Therefore, the new reward distribution may overlap with the old reward distribution. The oscillation of the learned reward function impedes the performance of the policy which is crucial in adversarial IRL setting.
>
> 5. t-test.
>
> We compared the best and second-best algorithms using a t-test. We ran the experiments 50 times for each seed and we tried 5 random seeds. In total, there are 250 experiment data points, and it should be enough to be considered as Gaussian distributions. The problem of different variances is solved by running a t-test bidirectionally. First, We run a t-test on population A with the mean of population B. Then we run a t-test on population B with the mean of population A. If both t-tests claim that population A outperforms population B then we say population A is better than population B.
>
> 6. Different seeds.
>
> We are not using the best policy among ALL 5 different seeds. We ran experiments on the best policy for EACH seed. Therefore, there are 5 best policies in total. Although selecting the best policy may favor unstable methods, it is unfair to select a non-converged policy to test.
>
> 7. Baselines for the noisy setting.
>
> Thanks for suggesting another baseline to test. We select EBIL as one of our baselines, which is designed to deal with the noisy demonstrations.
>
> 8. Learning curve and ablations.
>
> We will update our paper in the revised version.

---

> > ### Comment · Reviewer_uUoB · 2023-11-20
> >
> > You write that you have added an edge between states and actions. I am a bit confused why they should directly depend on each other. This is clearly different from the probabilistic model as it is in the control as inference framework. Could you clarify this for me?

---

> > ### Comment · Reviewer_PBAj · 2023-11-22
> > **Issues remain**
> >
> > Thank you for your reply. However, my raised concerns remain:
> >
> > 1. Eq. 10 (I think it was originally Eq. 9, although strangely openreview claims that there are no revisions) is wrong. $q(O_t|r_t)$ can not at the same time be proportional to $A(s,a)$ and equal to $\sigma(A)$. Even if it was supposed to mean $\propto \exp(A(s,a))$ (which it should), this would be very different from $\sigma(A)$.
> >
> > 2. B.2 States that $p(\mathcal{O}\_t=1 | E[r\_t]) = q(\mathcal{O}\_t | E[r\_t])$. This is in general not true, and in particular for the definition of q it is false. q can be any variational distribution, p refers to the true distribution. Hence, Theorem 2 is wrong: $q(\mathcal{O}_t | E[r_t])$ does not approximate the true distribution.
> >
> > 3. When $C(s,a)$ corresponds to the classifier output, then it does not approximate $p(\mathcal{O}|s,a)$. The optimal BCE output would approximate $\sigma(\log \frac{p\_{E}(s,a)}{p\_{\pi}(s,a)}) \neq p(\mathcal{O}|s,a)$
> >
> > 4. The advantage/Q-function involves an expectation over the next state no matter whether we are doing IRL or RL (where we also only have samples from the transition function).

---

> ### Author Response · Authors · 2023-11-21
>
> "This is clearly different from the probabilistic model as it is in the control as inference framework"
>
> You are correct. They should not depend on each other directly. We add the edge between $s_t$ and $a_t$ because we try to convince reviewer PBAj that the edge between $s_t$ and $a_t$ does not impact our theoretical result. The dependency between $s_t$ and $a_t$ is inferred by the forward and backward message.
>
> We have updated our paper with the original derivation and a new graphical model, in which we separate out the reward function node and make it a deterministic node.

---

### Official Review · Reviewer_uUoB · 2023-10-31

**Soundness:** 2 fair
**Presentation:** 3 good
**Contribution:** 3 good
**Rating:** 6
**Confidence:** 3

**Summary:**

This paper introduces a novel approach to IRL based on variational inference in a ''control as inference'' graphical model. For this, the authors formulate a graphical model with both reward and optimality random variables and derive a variational lower bound on trajectory log-likelihood. They discuss the validity of the approximate distribution of optimality and present an algorithm for maximizing the lower bound. The algorithm is evaluated in simulations across Mujoco environments and assistive gym.

**Strengths:**

I really enjoyed reading the paper as it offered me several new insights for approaching the IRL problem. The approach to tackle the IRL problem by means of variational inference in their graphical model seems novel and creative, and I can see wide applicability of the approach. Further, it may form a base for several new IRL methods, which are not restricted to the common maximum entropy formulation, so I believe that the paper could have a high impact on the community. The method's derivation through variational inference from the graphical model is notably elegant.

The paper is well-written, making even its technical content accessible. The approximation of the optimality distribution is discussed in sufficient detail, and the technical prerequisites are reasonable. While the paper lacks a demonstration of the tightness of the lower bound, it remains sufficient for a high-quality contribution.

In the evaluation, the algorithm reveals promising results, particularly in cases with sparse rewards.

**Weaknesses:**

The authors claim that its novelty lies in the "control as inference" graphical model. However, it is crucial to note that maximum-entropy-based IRL also hinges on inference within the "control as inference" graphical model. It can be derived as the maximum likelihood solution in this model and this is where the exponential distribution of the trajectory comes from (there are different derivations). Therefore, statements implying that the proposed method is distinct due to a lack of such modeling in IRL ("inspired by the graphical model for forward RL [...] and a general lack of such modeling in IRL") might be inaccurate. It would be beneficial for the paper to provide a more precise discussion of the differences between these probabilistic approaches.

The section on limitations and future work is notably brief and would benefit from more extensive exploration beyond the current concise treatment.

While I also like really like the "control as inference" review paper by Levine et al., this formulation predated their work and appears to stem from [1]. Given the mulitple explicit references to the control as inference paper, it would be appropriate to acknowledge the original work at some point.

__Minor:__

Equation 9: It should read $exp(A(s, a))$ instead of $A(s, a)$, I suppose?
Additionally, the equal sign in this equation should either be replaced with a proportional sign, or it should link the first and third term, i.e., $q(\ldots)$ and $\sigma(\ldots)$.

There are instances in the formulas where the bold font seems inconsistent. In Section 2.1, $s_t$ and $a_t$ are initially defined in bold, but this formatting is not maintained. If bold font indicates vectors, then the subscript $t$ should not be bold. This inconsistency also extends to the graphical model in Figure 1 (e.g., $O_t$ with bold $t$), and it impacted readability for me.

Between Equation 8 and 9, there is an extraneous period.

[1] Toussaint, M. (2009). Robot trajectory optimization using approximate inference. In Proceedings of the 26th annual international conference on machine learning (pp. 1049-1056).

**Questions:**

1. Why is it necessary to model the reward as random given the state and action? Would Equation 9 not remain unchanged if $q$ directly depended on the state and actions, thereby avoiding issues related to the gap between $p(O | s, a)$ and $p(O | r)$?

2. Is it necessary to retrain the classifier completely in each iteration? What is the computational cost associated with this process?

3. Do you have any insights into the potential sources of the increased performance compared to other IRL methods? It would be valuable to provide some explanation or analysis in this regard.

---

> ### Author Response · Authors · 2023-11-20
>
> Thank you for your response regarding our paper review. Below, you'll find our replies to your comments and feedback.
>
> 1.  Reference.
>
> We thank you for the advice on the relationship between max-ent IRL and the "control as inference" graphical model. We will update such a relationship in the updated version paper.
>
> 2.  Eqn.9.
>
> Thanks for pointing out the unclarity. We try to express that $q(...)$ is defined as $\sigma(...)$, which are the first and third terms.
>
> 3. The bold font.
>
> We have updated the bold state and action notations.
>
> 4. The necessity of modeling reward as random.
>
> The randomness of the reward function improves the performance of the algorithm in our experiments. We did not present this aspect in the paper. The assumption on why it improves the performance is the randomness makes the learned reward function change smoothly since the model optimizes both the mean and variance. Updating the mean usually leads to a higher variance. Therefore, the new reward distribution may overlap with the old reward distribution. The oscillation of the learned reward function impedes the performance of the policy which is crucial in adversarial IRL setting.
>
> 5. Classifier training details.
>
> We do not retrain the classifier completely. We use only one classifier and continue to train the classifier in each iteration, which aligns with GAIL and AIRL. In the experiments, the cost of training the classifier is super cheap compared to updating the policy and reward function.
>
> 6. Insights of the increased performance.
>
> The randomness of the reward function should be one of the reasons as I described in the second part of the response.

---

> > ### Comment · Reviewer_uUoB · 2023-11-20
> >
> > Thank you for your responses.
> >
> > Regarding 1: Could you elaborate on the probabilistic relationship between max-Ent IRL and your method? What are the differences? This would be an important point for my recommendation.
> >
> > Regarding 4: You write that "the randomness of the reward function improves the performance of the algorithm". How was the performance without the randomness in the tasks (also roughly in comparison to other IRL algorithms)? Furthermore, does it imply that you learn a full reward distribution? What is $R_\psi$ exactly in your implementation? If it is probabilistic, how do you deal with the randomness for updating the policy? I could not get these points from the paper.

---

> > > ### Author Response · Authors · 2023-11-21
> > >
> > > 1. Relationship between max-Ent IRL and our method.
> > >
> > > Max-Ent IRL utilizes a Boltzmann distribution to model demonstrations, in which the cost function $c_\phi$ determines the energy:
> > >
> > > \begin{align*}
> > > p_\phi(\tau)
> > > &= \frac{1}{Z}\exp(-c_\phi(\tau)),\\\\
> > > \end{align*}
> > >
> > > The estimation of the partition function $Z$ can be challenging, particularly in cases involving large or continuous domains, and it constitutes the primary computational hurdle. Ziebart et al. (2008) successfully compute $Z$ precisely through dynamic programming. Nonetheless, this approach is feasible only for small, discrete domains, and becomes infeasible when dealing with domains characterized by unknown system dynamics. Finn et al. (2016a) introduce an iterative sample-based method, i.e. guided cost learning, for estimating $Z$, and can scale to high-dimensional state and action spaces and nonlinear cost functions. Moreover, Finn et al. (2016b) have proved the equivalence between generative adversarial networks and guided cost learning. Our method is different from the above methods in the following aspects:
> > >
> > > a. Our method is unaware of the dynamics of the environment and can scale to high-dimensional state and action spaces and nonlinear reward function.
> > >
> > > b. Our method avoids computing the intractable partition function $Z$ by applying the structured variational inference. Precisely, the structure is to use $q(\mathcal{O}\_t|r_t)$ to approximate $p(\mathcal{O}\_t|s_t, a_t)$. Prior works focus on directly maximizing the $q(\mathcal{O}\_t|r_t)$ (they use the Boltzmann distribution described above), leading to unavoidable partition function Z. While we first use a learned classifier $C_\theta$, outputting a Bernoulli distribution indicating the probability of optimal state-action pair, then we use normalized reward or state value function (normalized by sigmoid function) to approximate this Bernoulli distribution.
> > >
> > >
> > > 2. Reward function
> > >
> > > Without the randomness, VLB-IRL never receives a positive cumulative reward in AssistiveGym environment, and its performance is almost consistent with IQ-Learn. For the Mojuco environment, VLB-IRL without randomness has a similar performance compared with VLB-IRL with randomness.
> > >
> > > "does it imply that you learn a full reward distribution?" Yes. $R_\phi$ outputs a Gaussian distribution by outputting the mean and std. In the paper, we use $R_\phi(s_t, a_t)$ to denote the Gaussian distribution.
> > >
> > >
> > > "how do you deal with the randomness for updating the policy?" For updating the learner policy, we let the learner policy interact with the environment and collect $s_t, a_t$. Then we send the $s_t, a_t$ to the reward model $R_\phi$ to get the distribution $R_\phi(s_t,a_t)$. Once we have the distribution $R_\phi(s_t, a_t)$, we use the reparameterization trick to sample a reward value $r_t$ from the Gaussian distribution and we use $r_t$ to update the learner policy.
> > >
> > > # References
> > >
> > > [1] Ziebart, Brian D., et al. "Maximum entropy inverse reinforcement learning." Aaai. Vol. 8. 2008.
> > >
> > > [2] Levine, Sergey. "Reinforcement learning and control as probabilistic inference: Tutorial and review." arXiv preprint arXiv:1805.00909 (2018).
> > >
> > > [3] Finn, Chelsea, et al. "A connection between generative adversarial networks, inverse reinforcement learning, and energy-based models." arXiv preprint arXiv:1611.03852 (2016).
> > >
> > > [4] Finn, Chelsea, Sergey Levine, and Pieter Abbeel. "Guided cost learning: Deep inverse optimal control via policy optimization." International conference on machine learning. PMLR, 2016.

---

> > > > ### Comment · Reviewer_uUoB · 2023-11-22
> > > >
> > > > When I wrote my initial review, I thought I did not understand minor details of the paper (randomness of the reward function and relations to maxEnt IRL). Unfortunately, these points seem to be larger issues. The authors' response could not clarify my uncertainty about the difference in the probabilistic model to maxEnt IRL, as they still do not take a "control as inference" perspective for maxEnt IRL. I think a comparison would be interesting and necessary to propose an alternative "control as inference" approach for IRL. If the authors are not familiar with this formulation, I recommend papers like [1] (chapter MCE IRL as maximum likelihood estimation) which found very useful to understand this better.
> > > > My wild guess would be that their problem formulation is exactly the same as maxEnt IRL if you remove the randomness of the reward function, but they take a different approach to solving it. This would also explain why the results for this case match those of the maxEnt methods. The probabilistic reward function could then make the results more robust, but this would need to be investigated further.
> > > >
> > > > The author's response could clarify my question on the probabilistic reward function.
> > > >
> > > > I still believe that the authors' approach is reasonable and could be useful, but a more developed theory in terms of the graphical model and relations with maxEnt IRL would be beneficial. There may also be further formal issues regarding the reward function and the VI derivation (discussions with other reviewers), which I could not check due to time reasons.
> > > >
> > > > [1] Gleave, Adam, and Sam Toyer. "A primer on maximum causal entropy inverse reinforcement learning." arXiv preprint arXiv:2203.11409 (2022).

---

### Official Review · Reviewer_CGqh · 2023-11-01

**Soundness:** 2 fair
**Presentation:** 3 good
**Contribution:** 2 fair
**Rating:** 3
**Confidence:** 4

**Summary:**

A new approach to imitation learning is presented, taking inspiration from the "control as inference" perspective. This method shares similarities with GAIL but introduces an additional step in the process. Instead of directly employing the discriminator to define the reward function, the proposed approach, known as VLB-IRL, trains a neural network to predict a Gaussian distribution over rewards when provided with state-action pairs. The policy is then updated based on rewards sampled from this distribution. The performance of this method is assessed in MuJoCo environments, where it demonstrates performance levels similar to established benchmarks like GAIL.

**Strengths:**

- This paper tried to theoretically explain the proposed method.
- The experimental protocol is well described.

**Weaknesses:**

- Graphical Model Clarification: The graphical model in the paper is quite unclear, particularly the representation of reward variables. Noting that the definition of V_t, representing the cumulative sum of rewards received after time t, is not adequately reflected in the graphical model. V_t captures the total reward received from time t onwards, and the reviewer suggests that this aspect should be more explicitly incorporated into the model for clarity.

- Dependence of Optimality on Reward: I am unsure about the rationale behind approximating optimality based on the reward. It is unclear whether having the reward information alone is sufficient to determine whether a state action pair is optimal. My question is whether the reward can serve as a reliable indicator of optimality and whether there is a clear justification for this approximation. This question is closely related to the following validity of the graphical model.

- Validity of Graphical Model: the entire theoretical development in the paper is based on the graphical model. They express concerns that if the graphical model's design is flawed or lacks clear justification, it could undermine the significance of the paper's contributions.

- Theoretical Results: The theoretical aspects of this paper appear to have certain limitations. Theorem 2, doesn't offer valuable insights into the convergence rate or the asymptotic behavior of the error bound. To derive Theorem 2, the proof mainly employs the property of a graphical model, however, the justification of the proposed graphical model is unclear, hence, it is not convincing.

**Questions:**

See weakness.

---

> ### Author Response · Authors · 2023-11-20
>
> Thank you for your response regarding our paper review. Below, you'll find our replies to your comments and feedback.
>
> 1. Graphical Model Clarification:
>
> $V_t$ is the value function. While cumulative rewards represent the actual sum of rewards an agent receives during an episode, the value function provides a more structured and general representation of the expected future rewards an agent can achieve from different states under a specific policy. The definition of the input of the value function only depends on the current state, which is $s_t$.
>
> 2. Dependence of Optimality on Reward and Validity of Graphical Model:
>
> We refer to Levine's paper (Reinforcement Learning and Control as Probabilistic Inference: Tutorial and Review). In the paper, the author has proved that with the optimality only depending on the reward, performing structured variational inference is equivalent to doing reinforcement learning in the max-ent setting. In our paper, we use exactly the same forward reinforcement learning, and the reward function is learned by our algorithm.
>
> 3. Theoretical Results:
>
> Theorem 2 claims that the approximation error is bounded by the variance of the reward function, which leads to deriving the objective function with the variance term.

---

### Meta-Review · Area_Chair_58Qi · 2023-12-05

**Metareview:**

This paper develops a new variational lower bound for inverse reinforcement learning based on a probabilistic graphical model. The reviewers found novelty in the high-level ideas of the paper. The experimental results were also a strength of the paper. However, there were multiple unresolved points of technical confusion that need to be fully addressed before the paper can be accepted for publication.

**Justification For Why Not Higher Score:**

Reviewer concerns about technical details and motivations were not resolved by the discussion with authors.

**Justification For Why Not Lower Score:**

N/A

---

### Decision · Program_Chairs · 2024-01-16

Reject